# Boosting Black Box Variational Inference

Francesco Locatello[*1,2], Gideon Dresdner[*2], Rajiv Khanna[3], Isabel Valera[1], and Gunnar Rätsch[2]

[1]Max-Planck Institute for Intelligent Systems, Germany
[2]Dept. for Computer Science, ETH Zurich, Universitätsstrasse 6, 8092 Zurich, Switzerland.
[3]The University of Texas at Austin, USA

## Abstract

Approximating a probability density in a tractable manner is a central task in Bayesian statistics. Variational Inference (VI) is a popular technique that achieves tractability by choosing a relatively simple variational approximation. Borrowing ideas from the classic boosting framework, recent approaches attempt to *boost* VI by replacing the selection of a single density with an iteratively constructed mixture of densities. In order to guarantee convergence, previous works impose stringent assumptions that require significant effort for practitioners. Specifically, they require a custom implementation of the greedy step (called the LMO) for every probabilistic model with respect to an unnatural variational family of truncated distributions. Our work fixes these issues with novel theoretical and algorithmic insights. On the theoretical side, we show that boosting VI satisfies a relaxed smoothness assumption which is sufficient for the convergence of the functional Frank-Wolfe (FW) algorithm. Furthermore, we rephrase the LMO problem and propose to maximize the Residual ELBO (RELBO) which replaces the standard ELBO optimization in VI. These theoretical enhancements allow for black box implementation of the boosting subroutine. Finally, we present a stopping criterion drawn from the duality gap in the classic FW analyses and exhaustive experiments to illustrate the usefulness of our theoretical and algorithmic contributions.

## 1 Introduction

Approximating probability densities is a core problem in Bayesian statistics, where inference translates to the computation of a posterior distribution. Posterior distributions depend on the modeling assumptions and can rarely be computed exactly. Variational Inference (VI) is a technique to approximate posterior distributions through optimization. It involves choosing a set of *tractable* densities, a.k.a. variational family, out of which the final approximation is to be chosen. The approximation is done by selecting a density in the candidate set that is close to the true posterior in terms of Kullback-Leibler (KL) divergence [1]. There is an inherent trade-off involved in fixing the set of tractable densities. Increasing the capacity of the variational family to approximate the posterior also increases the complexity of the optimization problem. Consider a degenerate case where the variational family contains just a single density. The optimization problem is trivial and runs in constant time, but the quality of the solution is poor and stands in no relation to the true posterior. This contrived example is clearly too restrictive, and in practice, the *mean field* approximation offers a good trade-off between expressivity and tractability [2]. However, in many real-world applications, mean field approximations are lacking in their ability to accurately approximate the posterior.

Imagine a practitioner that, after designing a Bayesian model and using a VI algorithm to approximate the posterior, finds that the approximation is too poor to be useful. Standard VI does not give the

---

[*]Authors contributed equally

practitioner the option to trade additional computational cost for a better approximation. As a result, there have been several efforts to ramp up the representative capacity of the variational family while maintaining tractability.

One line of work in this direction involves replacing the simple mean field family by a mixture of Gaussians. It is known that mixtures of Gaussians can approximate any distribution arbitrarily closely [18]. Boosting is a practical approach to finding the optimal approximating mixture and involves adding components to the mixture greedily one at a time [5, 13, 16]. Not only is boosting a practical solution, it also has well-studied trade-off bounds for number of iterations vs. approximation quality [13] by virtue of being essentially a variant of the classical Frank-Wolfe (FW) algorithm [8, 13]. Unfortunately, these greedy algorithms require a specialized, restricted variational family to ensure convergence and therefore a *white box* implementation of the boosting subroutine. These restrictions include that (a) each potential component of the mixture has a bounded support i.e., truncated densities, and (b) the subroutine should not return degenerate distributions. These assumptions require specialized care during implementation, and therefore, one cannot simply take existing VI solvers and boost them. This makes boosting VI unattractive for practitioners. In this work, we fix this issue by proposing a boosting *black box* VI algorithm that has many practical benefits.

Our work presents several key algorithmic and theoretical contributions, which we summarize below:

- We relax the previously known conditions for guaranteed convergence from smoothness to bounded curvature. As a consequence, the set of candidate densities no longer needs to be truncated, thereby easing its implementation and improving on the convergence guarantees.
- We propose a modified form of the ELBO optimization, the Residual ELBO (RELBO), which guarantees that the selected density is non-degenerate and is suitable for black box solvers (e.g. black box variational inference [19]).
- We propose a novel stopping criterion using the duality gap from FW, which is applicable to any boosting VI algorithm.

In addition to these theoretical contributions, we present extensive simulated and real-world empirical experiments to show the applicability of our proposed algorithmic extensions.

While our work is motivated by applications to VI, our theoretical results have a more general impact. We essentially analyze the application of the functional FW algorithm to the general task of minimizing the KL-divergence over a space of probability densities.

## 1.1 Related work

There is a vast literature on VI. We refer to [1] for a thorough review. Our focus is to use boosting to increase the complexity of a density, similar to the goal of Normalizing Flows [20], MCMC-VI hybrid methods [22, 21], or distribution transformations [23]. Our approach is in line with several previous approaches using mixtures of distributions to improve the expressiveness of the variational approximation [9, 6] but goes further to draw connections with classic optimization to obtain several novel theoretical and algorithmic insights.

While boosting has been well studied in other settings [15], it has only recently been applied to the problem of VI. Related works of [5] and [16] developed the algorithmic framework and conjectured a possible convergence rate of $O(1/T)$ but without theoretical analyses. The authors in [5] identify the need of truncated densities to ensure smoothness of the KL cost function. A more recent work [13] provides a theoretical base for analyzing the algorithm. They identify the sufficient conditions for guaranteeing convergence and provide explicit constants to the conjectured $O(1/T)$ rate. Unfortunately, these sufficient conditions amount to restrictive assumptions about the variational family and therefore require the practitioner to have white box access to the variational family and underlying VI algorithm. In this work, we remove these assumptions to allow use of black box VI methods.

Our analysis is mainly based on the FW algorithm [8], which is a commonly used algorithm for projection free constrained optimization. The convergence rates and requisite assumptions are well studied in various settings [12, 11, 14]. Its applications include non-Euclidean spaces, e.g., a variational objective for approximate marginal inference over the marginal polytope [10].

The rest of the paper is organized as follows. We begin by introducing and formalizing the boosting VI framework in Section 2. In Section 3, we review and analyze the Functional FW algorithm to greedily

solve the boosting VI. In Section 4, we first propose RELBO, an alternative of the contemporary ELBO optimization to implement a black box LMO (linear minimization oracle). Then, we propose a duality gap based stopping criterion for boosting VI algorithms. Finally, experimental evaluation is presented in Section 5. We refer the reader to the appendix for all proofs.

**Notation.** We represent vectors by small letters bold, (e.g. $\mathbf{x}$) and matrices by capital bold, (e.g. $\mathbf{X}$). For a non-empty subset $\mathcal{A}$ of some Hilbert space $\mathcal{H}$, let $\mathrm{conv}(\mathcal{A})$ denote its convex hull. $\mathcal{A}$ is often called *atom set* in the literature, and its elements are called *atoms*. The support of a density function $q$ is a measurable set denoted by capital letters sans serif i.e., Z. The inner product between two density functions $p, q : \mathsf{Z} \to \mathbb{R}$ in $L^2$ is defined as $\langle p, q \rangle := \int_{\mathsf{Z}} p(z)q(z)dz$.

## 2 Variational inference and boosting

Bayesian inference involves computing the posterior distribution given a model and the data. More formally, we choose a distribution for our observations $\mathbf{x}$ given unobserved latent variables $\mathbf{z}$, called the likelihood $p(\mathbf{x}|\mathbf{z})$, and a prior distribution over the latent variables, $p(\mathbf{z})$. Our goal is to infer the posterior, $p(\mathbf{z}|\mathbf{x})$ [1]. Bayes theorem relates these three distributions by expressing the posterior as equal to the product of prior and likelihood divided by the normalization constant, $p(\mathbf{x})$. The posterior is often intractable because the normalization constant $p(\mathbf{x}) = \int_{\mathsf{Z}} p(\mathbf{x}|\mathbf{z})p(\mathbf{z})d\mathbf{z}$ requires integrating over the full latent variable space.

The goal of VI is to find a *tractable* approximation $q(\mathbf{z})$ of $p(\mathbf{z}|\mathbf{x})$. From an optimization viewpoint, one can think of the posterior as an unknown function $p(\mathbf{z}|\mathbf{x}) : \mathsf{Z} \to \mathbb{R}^+_{>0}$ where Z is a measurable set. The task of VI is to find the best approximation, in terms of KL divergence, to this unknown function within a family of tractable distributions $\mathcal{Q}$. Therefore, VI can be written as the following optimization problem:

$$\min_{q(\mathbf{z}) \in \mathcal{Q}} D^{KL}(q(\mathbf{z})\|p(\mathbf{z}|\mathbf{x})). \tag{1}$$

It should be obvious that the quality of the approximation directly depends on the expressivity of the family $\mathcal{Q}$. However, as we increase the complexity of $\mathcal{Q}$, the optimization problem (1) also becomes more complex.

Rather than optimizing the objective (1) which requires access to an unknown function $p(\mathbf{z}|\mathbf{x})$ and is therefore not computable, VI equivalently maximizes instead the so-called Evidence Lower BOund (ELBO) [1]:

$$-\mathbb{E}_q\left[\log q(\mathbf{z})\right] + \mathbb{E}_q\left[\log p(\mathbf{x}, \mathbf{z})\right]. \tag{2}$$

A recent line of work [5, 16, 13] aims at replacing $\mathcal{Q}$ with $\mathrm{conv}(\mathcal{Q})$ thereby expanding the capacity of the variational approximation to the class of mixtures of the base family $\mathcal{Q}$:

$$\min_{q(\mathbf{z}) \in \mathrm{conv}(\mathcal{Q})} D^{KL}(q(\mathbf{z})\|p(\mathbf{x}, \mathbf{z})). \tag{3}$$

The boosting approach to this problem consists of specifying an iterative procedure, in which the problem (3) is solved via the greedy combination of solutions from simpler surrogate problems. This approach was first proposed in [5], and its connection to the FW algorithm was studied in [13]. Contrary to the boosting approaches in the deep generative modeling literature initiated by [24], boosting VI does not enjoy a simple and elegant subproblem as we discuss in Section 3.1. Next, we show how to tackle (3) from a formal and yet very practical optimization perspective.

## 3 Functional Frank-Wolfe for boosting variational inference

Taking a step back from the problem (3), we first define the general optimization problem and the relevant quantities needed for proving the convergence of FW. Then, we present the extension to boosting black box VI.

We start with the general optimization problem:

$$\min_{q \in \mathrm{conv}(\mathcal{A})} f(q). \tag{4}$$

where $\mathcal{A} \subset \mathcal{H}$ is closed and bounded and $f : \text{conv}(\mathcal{A}) \to \mathbb{R}$ is a convex functional with bounded curvature over its domain. Here the curvature is defined as in [8]:

$$C_{f,\mathcal{A}} := \sup_{\substack{s \in \mathcal{A}, \, q \in \text{conv}(\mathcal{A}) \\ \gamma \in [0,1] \\ y = q + \gamma(s-q)}} \frac{2}{\gamma^2} D(y,q), \tag{5}$$

where

$$D(y,q) := f(y) - f(q) - \langle y - q, \nabla f(q) \rangle.$$

It is known that if $\nabla f$ is Lipschitz continuous with constant $L$ (often referred to as $L$-smoothness) over $\text{conv}(\mathcal{A})$, then $C_{f,\mathcal{A}} \le L \, \text{diam}(\mathcal{A})^2$ where $\text{diam}(\mathcal{A}) := \max_{p,q \in \mathcal{A}} ||p - q||^2$ [8].

---

**Algorithm 1** Affine Invariant Frank-Wolfe

1: **init** $q^0 \in \text{conv}(\mathcal{A})$, $\mathcal{S} := \{q^0\}$, and accuracy $\delta > 0$
2: **for** $t = 0 \dots T$
3:      Find $s^t := (\text{Approx-})\text{LMO}_{\mathcal{A}}(\nabla f(q^t))$
4:      Variant 0: $\gamma = \frac{2}{\delta t + 2}$
5:      Variant 1: $\gamma = \arg\min_{\gamma \in [0,1]} f((1-\gamma)q^t + \gamma s^t)$
6:      $q^{t+1} := (1-\gamma)q^t + \gamma s^t$
7:      Variant 2: $\mathcal{S} = \mathcal{S} \cup s^t$
8:          $q^{t+1} = \arg\min_{q \in \text{conv}(\mathcal{S})} f(q)$
9: **end for**

---

The FW algorithm is depicted in Algorithm 1. Note that Algorithm 2 in [5] is a variant of Algorithm 1. In each iteration, the FW algorithm queries a so-called Linear Minimization Oracle (LMO) which solves the following inner optimization problem:

$$\text{LMO}_{\mathcal{A}}(y) := \arg\min_{s \in \mathcal{A}} \langle y, s \rangle, \tag{6}$$

for a given $y \in \mathcal{H}$ and $\mathcal{A} \subset \mathcal{H}$. To tackle the constrained convex minimization problem in Eq. (4), Frank-Wolfe iteratively solves a linear constrained problem where, at iteration $t$, the function $f$ is replaced by its first-order Taylor approximation around the current iterate $q^t$. It is easy to see that the solution of this problem can be obtained by querying $\text{LMO}(\nabla f(q^t))$. Indeed, the following holds: $\arg\min_{s \in \text{conv}(\mathcal{A})} f(q^t) + \langle \nabla f(q^t), s - q^t \rangle = \arg\min_{s \in \text{conv}(\mathcal{A})} \langle \nabla f(q^t), s \rangle =: \text{LMO}(\nabla f(q^t))$. Depending on $\mathcal{A}$, computing an exact solution of (6) can be hard in practice. This motivates the approximate LMO which returns an approximate minimizer $\tilde{s}$ of (6) for some accuracy parameter $\delta$ and the current iterate $q^t$ such that:

$$\langle y, \tilde{s} - q^t \rangle \le \delta \min_{s \in \mathcal{A}} \langle y, s - q^t \rangle \tag{7}$$

We discuss a simple algorithm to implement the LMO in Section 4. Finally, we find that Algorithm 1 is known to converge sublinearly to the minimizer $q^\star$ of (4) with the following rate:

**Theorem 1** ([8]). *Let $\mathcal{A} \subset \mathcal{H}$ be a closed and bounded set and let $f : \mathcal{H} \to \mathbb{R}$ be a convex function with bounded curvature $C_{f,\mathcal{A}}$ over $\mathcal{A}$. Then, the Affine Invariant FW algorithm (Algorithm 1) converges for $t \ge 0$ as*

$$f(q^t) - f(q^\star) \le \frac{2\left(\frac{1}{\delta} C_{f,\mathcal{A}} + \varepsilon_0\right)}{\delta t + 2}$$

*where $\varepsilon_0 := f(q^0) - f(q^\star)$ is the initial error in objective, and $\delta \in (0,1]$ is the accuracy parameter of the approximate LMO.*

**Discussion.** Theorem 1 has several implications for boosting VI. First, the LMO problem does not need to be solved exactly in order to guarantee convergence. Second, Theorem 1 guarantees that Algorithm 1 converges to the best approximation in $\text{conv}(\mathcal{A})$ which, depending on the expressivity of the base family, could even contain the full posterior. Furthermore, the theorem gives a convergence rate which states that, in order to achieve an error of $\epsilon$, we need to perform $O(\frac{1}{\epsilon})$ iterations.

Similar discussions are also presented by [13]. The crucial question, which they leave unaddressed, is whether under their assumptions there exists a variational family of densities which (a) is expressive enough to well-approximate the posterior; (b) satisfies the conditions required to guarantee convergence; and (c) allows for efficient implementation of the LMO.

### 3.1 Curvature of boosting variational inference

In order to boost VI using FW in practice, we need to ensure that the assumptions are not violated. Assume that $\mathcal{A} \subset \mathcal{Q}$ is the set of probability density functions with compact parameter space as well

as bounded infinity norm and $L2$ norm. These assumptions on the search space are easily justified since it is reasonable to assume that the posterior is not degenerate (bounded infinity norm) and has modes that are not arbitrarily far away from each other (compactness). Under these assumptions, the optimization domain is closed and bounded. It is simple to show that the solution of the LMO problem over $\mathrm{conv}(\mathcal{A})$ is an element of $\mathcal{A}$. Therefore, $\mathcal{A}$ is closed. The troublesome condition that needs to be satisfied for the convergence of FW is smoothness. Indeed, the work of [5] already recognized that the boosting literature typically require a smooth objective and showed that densities bounded away from zero are sufficient. [13] showed that the necessary condition to achieve smoothness is that the approximation be not arbitrarily far from the optimum. They argue that while this is a practical assumption, the general theory would require truncated densities. We relax this assumption. As per Theorem 1, a bounded curvature is actually sufficient to guarantee convergence. This condition is weaker than smoothness, which was assumed by [13, 5]. For the KL divergence, the following holds.

**Theorem 2.** *$C_{f,\mathcal{A}}$ is bounded for the KL divergence if the parameter space of the densities in $\mathcal{A}$ is bounded.*

The proof is provided in Appendix A.

**Discussion.** Surprisingly, a bounded curvature for the $D^{KL}$ can be obtained as long as:

$$\sup_{\substack{s \in \mathcal{A},\, q \in \mathrm{conv}(\mathcal{A}) \\ \gamma \in [0,1] \\ y = q + \gamma(s-q)}} \frac{2}{\gamma^2} D^{KL}(y \| q)$$

is bounded. The proof sketch proceeds as follows. For any pair $s$ and $q$, we need to check that $\frac{2}{\gamma^2} D^{KL}(y\|q)$ is bounded as a function of $\gamma \in [0,1]$. The two limit points, $D^{KL}(s\|q)$ for $\gamma = 1$ and $\|s-q\|_2^2$ for $\gamma = 0$, are both bounded for any choice of $s$ and $q$. Hence, the $C_{f,\mathcal{A}}$ is bounded as it is a continuous function of $\gamma$ in $[0,1]$ with bounded function values at the extreme points. $D^{KL}(s\|q)$ is bounded because the parameter space is bounded. $\|s-q\|_2^2$ is bounded by the triangle inequality and bounded $L2$ norm of the elements of $\mathcal{A}$. This result is particularly relevant, as it makes the complicated truncation described in [13] unnecessary without any additional assumption. Indeed, while a bounded parameter space was already assumed in [13] and is a practical assumption, truncation is tedious to implement. Note that [5] considers full densities as an approximation of the truncated one. They also argue that the theoretically grounded family of distributions for boosting should contain truncated densities. Avoiding truncation has another very important consequence for the optimization. Indeed, [13] proves convergence of boosting VI only to a truncated version of the posterior. Therefore, Theorem 8 in [13] contains a term that does not decrease when the number of iteration increases. While this term could be small, as it contains the error of the approximation on the tails, it introduces a crucial hyperparameter in the algorithm i.e., where to truncate. Instead, we here show that under much weaker assumptions on the set $\mathcal{A}$, it is possible to converge to the true posterior.

# 4 The residual ELBO: implementing a black box LMO

Note that the LMO is a constrained linear problem in a function space. A complicated heuristic is developed in [5] to deal with the fact that the *unconstrained* linear problem they consider has a degenerate solution. The authors of [13] propose to use projected stochastic gradient descent on the parameters of $s$. The problem with this is that, to the best of our knowledge, the convergence of projected stochastic gradient descent is not yet understood in this setting. To guarantee convergence of the FW procedure, it is crucial to make sure that the solution of the LMO is not a degenerate distribution. This translates to a constraint on the infinity norm of $s$. Such a constraint is hardly practical. Indeed, one must be able to compute the maximum value of $s$ as a function of its parameters which depends on the particular choice of $\mathcal{A}$. In contrast, the entropy is a general term that can be approximated via sampling and therefore allows for black box computation. We relate infinity norm and entropy in the following lemma.

**Lemma 3.** *A density with bounded infinity norm has entropy bounded from below. The converse is true for many of the distributions which are commonly used in VI (for example Gaussian, Cauchy and Laplace).*

The proof is provided in Appendix A.

In general, bounded entropy does not always imply bounded infinity norm. While this is precisely the statement we need, a simple verification is sufficient to show that it holds in most cases of interest. We assume that $\mathcal{A}$ is a family for which bounded entropy implies bounded infinity norm. Therefore, we can constrain the optimization problem with the entropy instead of the infinity norm. We call $\bar{\mathcal{A}}$ the family $\mathcal{A}$ without the infinity norm constraint. At every iteration, we need to solve:

$$\underset{\substack{s \in \bar{\mathcal{A}} \\ H(s) \geq -M}}{\arg\min} \left\langle s, \log\left(\frac{q^t}{p}\right) \right\rangle$$

Note that this is simply the LMO from Equation (6) with $y = \nabla_q D^{KL}(q^t \| p) = \log \frac{q^t}{p}$ but with an additional constraint on the entropy. This constraint on the entropy is crucial since otherwise the solution of the LMO would be a degenerate distribution as the authors of [5] have also argued.

We now replace this problem with its regularized form using Lagrange multipliers and solve for $s$ given a fixed value of $\lambda$:

$$\underset{s \in \bar{\mathcal{A}}}{\arg\min} \left\langle s, \log\left(\frac{q^t}{p}\right) \right\rangle + \lambda\left(-H(s) - M\right) = \underset{s \in \bar{\mathcal{A}}}{\arg\min} \left\langle s, \log\left(\frac{q^t}{p}\right) \right\rangle + \langle s, \log s^\lambda \rangle \quad (8)$$

$$= \underset{s \in \bar{\mathcal{A}}}{\arg\min} \left\langle s, \log\left(\frac{s^\lambda}{\frac{p}{q^t}}\right) \right\rangle$$

$$= \underset{s \in \bar{\mathcal{A}}}{\arg\min} \left\langle s, \log\left(\frac{s}{\sqrt[\lambda]{\frac{p}{q^t}}}\right) \right\rangle.$$

Therefore, the regularized LMO problem is equivalent to the following minimization problem:

$$\underset{s \in \bar{\mathcal{A}}}{\arg\min} \, D^{KL}(s \| \frac{1}{Z} \sqrt[\lambda]{\frac{p}{q^t}}),$$

where $Z$ is the normalization constant of $\sqrt[\lambda]{\frac{p}{q^t}}$. From this optimization problem, we can write what we call the Residual Evidence Lower Bound (RELBO) as:

$$\text{RELBO}(s, \lambda) := \mathbb{E}_s[\log p] - \lambda \mathbb{E}_s[\log s] - \mathbb{E}_s[\log q^t]. \quad (9)$$

**Discussion.** Let us now analyze the RELBO and compare it with the ELBO in standard VI [1]. First, note that we introduce the hyperparameter $\lambda$ which controls the weight of the entropy. In order to obtain the true LMO solution, one would need to maximize the LHS of Equation (8) for $\lambda$ and solve the saddle point problem. In light of the fact that an approximate solution is sufficient for convergence, we consider the regularized problem as a simple heuristic. One can then fix an arbitrary value for $\lambda$ or decrease it when $t$ increases. The latter amounts to allowing increasingly sharp densities as optimization proceeds. The other important difference between ELBO and RELBO is the residual term which is expressed through $\mathbb{E}_s[\log p] - \mathbb{E}_s[\log q^t]$. Maximizing this term amounts to looking for a density with low cross entropy with the joint $p$ and high cross entropy with the current iterate $q^t$. In other words, the next component $s^t$ needs to be as close as possible to the target $p$ but also sufficiently different from the current approximation $q^t$. Indeed, $s^t$ should capture the aspects of the posterior that the current mixture could not approximate yet.

**Failure Modes.** Using a black box VI as an implementation for the LMO represents an attractive practical solution. Indeed, one could just run VI once and, if the result is not good enough, run it again on the residual without changing the structure of the implementation. Unfortunately, there are two failure modes that should be discussed. First, if the target posterior is a perfectly symmetric multimodal distribution, then the residual is also symmetric and the algorithm may get stuck. A simple solution to this problem is to run the black box VI for fewer iterations, breaking the symmetry of the residual. The second problem arises in scenarios where the posterior distribution can be approximated well by a single element of $\mathcal{Q}$. In such cases, most of the residual will be on the tails. The algorithm will then fit the tails and in the following iterations re-learn a distribution close to $q^0$. As a consequence, it is important to identify good solutions before investing additional computational effort by adding more components to the mixture. Note that the ELBO cannot be used for this purpose, as its value at the maximum is unknown.

**Stopping criterion.** We propose a stopping criterion for boosting VI which allows us to identify when a reasonably good approximation is reached and save computational effort. To this end, we rephrase the notion of *duality gap* [8, 7] in the context of boosting VI, which gives a surprisingly simple stopping criterion for the algorithm.

**Lemma 4.** *The duality gap* $g(q) := \max_{s \in \text{conv}(\mathcal{A})} \langle q - s, \log \frac{q}{p} \rangle$ *computed at some iterate* $q \in \text{conv}(\mathcal{A})$ *is an upper bound on the primal error* $D^{KL}(q\|p) - D^{KL}(q^\star\|p)$.

The proof is provided in Appendix A.

Note that the $\arg\max_{s \in \text{conv}(\mathcal{A})} \langle q - s, \log \frac{q}{p} \rangle$ is precisely the LMO solution to the problem (6). Therefore, with an exact LMO, one obtains a certificate on the primal error for free, without knowing the value of $D^{KL}(q^\star\|p)$. It is possible to show that a convergence rate similar to Theorem 1 also holds for the duality gap [8]. If the oracle is inexact, the estimate of the duality gap $\tilde{g}(q)$ satisfies that $\frac{1}{\delta}\tilde{g}(q) \geq g(q)$, as a consequence of (7).

# 5 Experimental evaluation

Notably, our VI algorithm is black box in the sense that it leaves the definition of the model and the choice of variational family up to the user. Therefore, we are able to reuse the same boosting black box VI solver to run all our experiments, and more generally, any probabilistic model and choice of variational family. We chose to implement our algorithm as an extension to the *Edward* probabilistic programming framework [25] thereby enabling users to apply boosting VI to any probabilistic model and variational family which are definable in *Edward*. In Appendix B, we show a code sample of our implementation of Bayesian logistic regression.

For comparisons to baseline VI, we use *Edward*'s built-in black box VI (BBVI) algorithm without modification. We run these baseline VI experiments for 10,000 iterations which is orders of magnitude more than what is required for convergence. Unless otherwise noted, we use Gaussians as our base family. Note that FW with fixed step size is not monotonic and so in the experiments in which we use a fixed step size, it is expected that the last iteration is not optimal. We use the training log-likelihood to select the best iteration and we used the duality gap as a diagnostic tool in the implementation to understand the impact of $\lambda$. We found that $\lambda = \frac{1}{\sqrt{t+1}}$ worked well in all the experiments. Code to reproduce the experiments is available at: https://github.com/ratschlab/boosting-bbvi.

## 5.1 Synthetic data

First, we use synthetic data to visualize the approximation of our algorithm of a bimodal posterior. In particular, we consider a mixture of two Gaussians with parameters $\mu = (-1, +1)$, $\sigma = (0.5, 0.5)$, and mixing weights $\pi = (0.4, 0.6)$.

We performed experiments with all three variants of FW described in Algorithm 1. For the fully corrective variant, we used FW to solve the subproblem of finding the optimal weights for the current atom set. We observe that unlike BBVI, all three variants are able to fit both modes of the bimodal target distribution. The fully corrective version gives the best fit. Unfortunately, this improved solution comes at a computational cost — solving the line search and fully corrective subproblems is dramatically

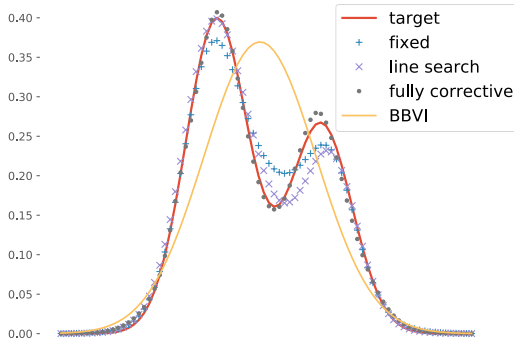

Figure 1: Comparison between BBVI and three variants of our boosting BBVI method on a mixture of Gaussians example.

slower than the fixed step size variant. In the experiments that follow we were able to improve upon the initial VI solution using the simple fixed step size. We believe this is the most interesting variant for practitioners as it does not require any additional implementation other than the VI subroutine. Our synthetic data results are summarized in Figure 1.

Table 1: Comparison of boosting BBVI on the CHEMREACT dataset. We observe that using the Laplace distribution as the base family, our method outperforms BBVI using either Laplace or Gaussian distributions as the variational family. In addition, boosting BBVI has lower variance across repetitions.

| | Train LL | Test AUROC |
|---|---|---|
| Boosting BBVI (Laplace) | -.677 ± 0.002 | **0.794 ± 0.005** |
| BBVI Edward (Laplace) | -0.681 ± 0.003 | 0.781 ± 0.012 |
| BBVI Edward (Gaussian) | -0.671 ± 0.002 | 0.790 ± 0.009 |
| Line Search Boosting VI ([5]) | -2.808 | 0.6377 |
| Fixed Step Boosting VI ([13]) | -3.045 | 0.6193 |
| Norm Corrective Boosting VI ([13]) | -2.725 | 0.6440 |

Table 2: Comparison of boosting BBVI on EICU COLLABORATIVE RESEARCH dataset. We observe that our method outperforms BBVI with the Laplace distribution. In addition, boosting BBVI has lower variance across repetitions.

| | Train LL | Test AUROC |
|---|---|---|
| Boosting BBVI (Laplace) | **-0.195 ± 0.007** | **0.844 ± 0.006** |
| BBVI Edward (Laplace) | -0.200 ± 0.032 | 0.838 ± 0.016 |

Table 3: Matrix factorization results for latent dimension $D = 3, 5, 10$ on the CBCL FACE dataset. We observe that our method outperforms the baseline BBVI method on mean-squared error (MSE).

| | BBVI MSE | Boosting BBVI MSE | BBVI Test LL | Boosting BBVI Test LL |
|---|---|---|---|---|
| D=3 | 0.0184 ± 0.001 | **0.0139 ± 0.44e-04** | -0.9363 ± 0.6e-3 | **-0.9354 ± 0.3e-3** |
| D=5 | 0.0187 ± 0.001 | **0.0137 ± 0.53e-04** | **-0.9391 ± 0.6e-3** | -0.9393 ± .4e-3 |
| D=10 | 0.0188 ± 0.001 | **0.0135 ± 0.52e-04** | **-0.9468 ± 0.3e-3** | -0.9492 ± .001 |

## 5.2 Bayesian logistic regression on two real-world datasets

In this experiment, we consider two real-world binary-classification tasks: predicting the reactivity of a chemical and predicting mortality in the intensive case unit (ICU). For both tasks we use the Bayesian logistic regression model. This allows us to compare to previous work in [13]. Bayesian logistic regression is a conditional prediction model with prior $p(\mathbf{w}) = \mathcal{N}(0, 1)$ on the weights and conditional likelihood $p(\mathbf{y}|\mathbf{X}) = \text{Bernoulli}(p = \text{sigmoid}(\mathbf{X}^\top \mathbf{w}))$. This model is commonly used as an example of a simple model which does not have a closed form posterior. [1]

We use the CHEMREACT dataset which contains 26,733 chemicals, each with 100 features. For this experiment, we ran our algorithm for 35 iterations and found that iteration 17 had the best performance. We observe that running merely one single well-tuned iteration of BBVI as implemented in the *Edward* framework using Gaussian as the variational class outperforms 10 iterations of boosting VI in [13]. While BBVI already has good performance in terms of AUROC, we are able to improve it further by using the fixed step size variant of FW and the Laplace distributions as the base family. In addition, our solution is more stable, namely it has lower standard deviation across replications. Results are summarized in Table 1.

For the mortality prediction task, we used a preprocessed dataset created by the authors of [3] from the EICU COLLABORATIVE RESEARCH database [4]. The preprocessing included selecting patient stays between 1 and 30 days, removing patients with missing values, and selecting a subset of clinically relevant features. The final dataset contains 71,366 patient stays and 70 relevant features ranging from age and gender to lab test results. We performed a 70-30% train-test split. We ran our algorithm for 29 iterations and again found that iteration 17 had the best performance. We observed that our method improves upon the AUROC of *Edward*'s baseline VI solution and is also more stable. Results are summarized in Table 2.

## 5.3 Bayesian matrix factorization

Bayesian Matrix Factorization [17] is a more complex model defined in terms of two latent variables, $\mathbf{U}$ and $\mathbf{V}$ for some choice of the latent dimension $D$. In the base distribution, each entry of the matrices $\mathbf{U}$ and $\mathbf{V}$ are independent Gaussians. To sample from $\mathbf{R}^t$, we sample $U, V$ from the

boosted posterior $(\mathbf{U}, \mathbf{V})^t$ and then sample from $\mathcal{N}(U^\top V, I)$. Thus, $\mathbf{R}^t \sim \mathcal{N}(\mathbf{U}^{t^\top} \mathbf{V}^t, I)$ where $(\mathbf{U}, \mathbf{V})^t \sim \sum_i^t \alpha^i s^i(\mathbf{U}, \mathbf{V})$ and $s^i(\mathbf{U}, \mathbf{V})$ is the $t$-th iterate returned by the LMO.

We use the CBCL FACE[1] dataset which is composed of 2,492 images of 361 pixels each, arranged into a matrix. Using a 50% mask for the train-test split, we performed matrix completion on this data using the above model. We compared our boosting BBVI to BBVI for three choices of the latent dimension $D = 3, 5, 10$ and observe improvements across all three in mean-squared error. For held-out test log-likelihood, we observe an improved performance for $D = 3$. Similar to the results for Bayesian linear regression, we observe that the variance across replications is also smaller for our method. Surprisingly, increasing $D$ does not have a significant effect on either of the metrics which may indicate that a relatively inexpressive model ($D = 3$) already contains a good approximation. Results are summarized in Table 3.

## 6 Conclusion

We have presented a refined yet practical theoretical analysis for the boosting variational inference paradigm. Our approach incorporates black box VI solvers into a general gradient boosting framework based on the Frank-Wolfe algorithm. Furthermore, we introduced a subroutine which is finally attractive to practitioners as it does not require any additional overhead beyond running a general black box VI solver multiple times. This is an important step forward in adding boosting VI to the standard toolbox of Bayesian inference.

**Acknowledgements.** FL is partially supported by the Max-Planck ETH Center for Learning Systems. FL, GD are partially supported by an ETH core grant (to GR). RK is supported by NSF Grant IIS 1421729. We thank Matthias Hüser for providing the preprocessed eICU dataset. We also thank David Blei and Anant Raj for helpful discussions.

## Footnotes

[1] http://cbcl.mit.edu/software-datasets/FaceData2.html

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
