[Supplementary Material]

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

=3 | 0.0184 $\pm$ 0.001 | **0.0139 $\pm$ 0.44e-04** | -0.9363 $\pm$ 0.6e-3 | **-0.9354 $\pm$ 0.3e-3** |
| D=5 | 0.0187 $\pm$ 0.001 | **0.0137 $\pm$ 0.53e-04** | **-0.9391 $\pm$ 0.6e-3** | -0.9393 $\pm$ .4e-3 |
| D=10 | 0.0188 $\pm$ 0.001 | **0.0135 $\pm$ 0.52e-04** | **-0.9468 $\pm$ 0.3e-3** | -0.9492 $\pm$ .001 |

## 5.2 Bayesian logistic regression on two real-world datasets

In this experiment, we consider two real-world binary-classification tasks: predicting the reactivity of a chemical and predicting mortality in the intensive case unit (ICU). For both tasks we use the Bayesian logistic regression model. This allows us to compare to previous work in [13]. Bayesian logistic regression is a conditional prediction model with prior $p(\mathbf{w}) = \mathcal{N}(0, 1)$ on the weights and conditional likelihood $p(\mathbf{y}|\mathbf{X}) = \text{Bernoulli}(p = \text{sigmoid}(\mathbf{X}^\top \mathbf{w}))$. This model is commonly used as an example of a simple model which does not have a closed form posterior. [1]

We use the CHEMREACT dataset which contains 26,733 chemicals, each with 100 features. For this experiment, we ran our algorithm for 35 iterations and found that iteration 17 had the best performance. We observe that running merely one single well-tuned iteration of BBVI as implemented in the *Edward* framework using Gaussian as the variational class outperforms 10 iterations of boosting VI in [13]. While BBVI already has good performance in terms of AUROC, we are able to improve it further by using the fixed step size variant of FW and the Laplace distributions as the base family. In addition, our solution is more stable, namely it has lower standard deviation across replications. Results are summarized in Table 1.

For the mortality prediction task, we used a preprocessed dataset created by the authors of [3] from the EICU COLLABORATIVE RESEARCH database [4]. The preprocessing included selecting patient stays between 1 and 30 days, removing patients with missing values, and selecting a subset of clinically relevant features. The final dataset contains 71,366 patient stays and 70 relevant features ranging from age and gender to lab test results. We performed a 70-30% train-test split. We ran our algorithm for 29 iterations and again found that iteration 17 had the best performance. We observed that our method improves upon the AUROC of *Edward*'s baseline VI solution and is also more stable. Results are summarized in Table 2.

## 5.3 Bayesian matrix factorization

Bayesian Matrix Factorization [17] is a more complex model defined in terms of two latent variables, $\mathbf{U}$ and $\mathbf{V}$ for some choice of the latent dimension $D$. In the base distribution, each entry of the matrices $\mathbf{U}$ and $\mathbf{V}$ are independent Gaussians. To sample from $\mathbf{R}^t$, we sample $U, V$ from the

boosted posterior $(\mathbf{U}, \mathbf{V})^t$ and then sample from $\mathcal{N}(U^\top V, I)$. Thus, $\mathbf{R}^t \sim \mathcal{N}(\mathbf{U}^{t\top}\mathbf{V}^t, I)$ where $(\mathbf{U}, \mathbf{V})^t \sim \sum_i^t \alpha^i s^i(\mathbf{U}, \mathbf{V})$ and $s^i(\mathbf{U}, \mathbf{V})$ is the $t$-th iterate returned by the LMO.

We use the CBCL FACE[1] dataset which is composed of 2,492 images of 361 pixels each, arranged into a matrix. Using a 50% mask for the train-test split, we performed matrix completion on this data using the above model. We compared our boosting BBVI to BBVI for three choices of the latent dimension $D = 3, 5, 10$ and observe improvements across all three in mean-squared error. For held-out test log-likelihood, we observe an improved performance for $D = 3$. Similar to the results for Bayesian linear regression, we observe that the variance across replications is also smaller for our method. Surprisingly, increasing $D$ does not have a significant effect on either of the metrics which may indicate that a relatively inexpressive model ($D = 3$) already contains a good approximation. Results are summarized in Table 3.

## 6  Conclusion

We have presented a refined yet practical theoretical analysis for the boosting variational inference paradigm. Our approach incorporates black box VI solvers into a general gradient boosting framework based on the Frank-Wolfe algorithm. Furthermore, we introduced a subroutine which is finally attractive to practitioners as it does not require any additional overhead beyond running a general black box VI solver multiple times. This is an important step forward in adding boosting VI to the standard toolbox of Bayesian inference.

**Acknowledgements.** FL is partially supported by the Max-Planck ETH Center for Learning Systems. FL, GD are partially supported by an ETH core grant (to GR). RK is supported by NSF Grant IIS 1421729. We thank Matthias Hüser for providing the preprocessed eICU dataset. We also thank David Blei and Anant Raj for helpful discussions.

## Footnotes

[1]http://cbcl.mit.edu/software-datasets/FaceData2.html

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

## A    Proof of the Main Results

**Lemma' 3.** *A density with bounded infinity norm has entropy bounded from below.*

*Proof.* Assume the infinity norm of a density $s$ is bounded from above by a constant $M$. This implies that $s(x) \leq M \; \forall \; x$. Therefore:

$$-H(s) \leq |H(s)|$$

$$\leq \int_{\mathsf{X}} |s(x)| \cdot |\log(s(x))| \, dx$$

$$\leq \int_{\mathsf{X}} |s(x)| \cdot |\log(M)| \, dx$$

$$= \int_{\mathsf{X}} s(x) \cdot |\log(M)| \, dx$$

$$= |\log(M)|$$

Therefore $H(s) \geq -|\log(M)|$ which concludes the proof. $\qquad\square$

**Theorem' 2.** *$C_{f,\mathcal{A}}$ is bounded for the KL divergence if the parameters space of the densities in $\mathcal{A}$ is bounded.*

*Proof.*

$$D(y,q) = D^{KL}(y) - D^{KL}(q) - \langle y - q, \nabla D^{KL}(q) \rangle$$

$$= \langle y, \log \frac{y}{p} \rangle - \langle q, \log \frac{q}{p} \rangle - \langle y - q, \log \frac{q}{p} \rangle$$

$$= \langle y, \log \frac{y}{p} \rangle - \langle y, \log \frac{q}{p} \rangle$$

$$= \langle y, \log \frac{y}{q} \rangle$$

$$= D^{KL}(y\|q)$$

In order to show that $C_{f,\mathcal{A}}$ is bounded we then need to show that:

$$\sup_{\substack{s\in\mathcal{A}, \, q\in\mathrm{conv}(\mathcal{A}) \\ \gamma\in[0,1] \\ y=q+\gamma(s-q)}} \frac{2}{\gamma^2} D^{KL}(y\|q)$$

is bounded. For a fixed $s$ and $q$ we how that $\frac{2}{\gamma^2} D^{KL}(y\|q)$ is continuous. Since the parameter space is bounded $D^{KL}(y\|q)$ is always bounded for any $\gamma \geq \varepsilon > 0$ and so is the $C_{f,\mathcal{A}}$, therefore the $C_{f,\mathcal{A}}$ is continuous for $\gamma \in (0,1]$. We only need to show that it also holds for $\gamma = 0$ in order to use the result that a continuous function on a bounded domain is bounded. When $\gamma \to 0$ we have that both $\gamma^2$ and $D^{KL}(y\|q) \to 0$. Therefore we use L'Hospital Rule (H) and obtain:

$$\lim_{\gamma\to 0} \frac{2}{\gamma^2} D^{KL}(y\|q) \overset{H}{=} \lim_{\gamma\to 0} \frac{1}{\gamma} \int_{\mathsf{X}} (s-q) \log\left(\frac{y}{q}\right)$$

$$\lim_{\gamma\to 0} \frac{2}{\gamma^2} D^{KL}(y\|q) \overset{H}{=} \lim_{\gamma\to 0} \frac{1}{\gamma} \int_{\mathsf{X}} (s-q) \log\left(\frac{y}{q}\right)$$

where for the derivative of the $D^{KL}$ we used the functional chain rule. Again both numerator and denominators in the limit go to zero when $\gamma \to 0$, so we use L'Hospital Rule again and obtain:

$$\lim_{\gamma\to 0} \frac{1}{\gamma} \int_{\mathsf{X}} (s-q) \log\left(\frac{y}{q}\right) \overset{H}{=} \lim_{\gamma\to 0} \int_{\mathsf{X}} (s-q)^2 \frac{q}{y}$$

$$= \int_{\mathsf{X}} \lim_{\gamma\to 0} (s-q)^2 \frac{q}{y}$$

$$= \int_{\mathsf{X}} (s-q)^2$$

which is bounded under the assumption of bounded parameters space and bounded infinity norm. Indeed:

$$\int_{\mathsf{X}} (s - q)^2 \leq 4 \max_{s \in \mathrm{conv}(\mathcal{A})} \int_{\mathsf{X}} s^2$$

Which is bounded under the assumption of bounded $L2$ norm of the densities in $\mathcal{A}$ by triangle inequality. $\qquad\square$

**Lemma' 4.** *The duality gap* $g(q) := \max_{s \in \mathrm{conv}(\mathcal{A})} \langle q - s, \log \frac{q}{p} \rangle$ *computed at some iterate* $q \in \mathrm{conv}(\mathcal{A})$ *is an upper bound on the primal error* $D^{KL}(q\|p) - D^{KL}(q^\star\|p)$.

*Proof.* $\log \frac{q}{p}$ is the gradient of the $D^{KL}(q\|p)$ computed at some $q \in \mathrm{conv}(\mathcal{Q})$. Therefore, the dual function ([7, Section 2.2]) of the $D^{KL}$ is:

$$w(q) := \min_{s \in \mathrm{conv}(\mathcal{A})} D^{KL}(q\|p) + \langle s - q, \log \frac{q}{p} \rangle.$$

By definition, the gradient is a linear approximation to a function lying below its graph at any point. Therefore, we have that for any $q, y \in \mathrm{conv}(\mathcal{A})$:

$$w(q) = \min_{s \in \mathrm{conv}(\mathcal{A})} D^{KL}(q\|p) + \langle s - q, \log \frac{q}{p} \rangle \leq D^{KL}(q\|p) + \langle y - q, \log \frac{q}{p} \rangle \leq D^{KL}(y\|p).$$

The duality gap at some point $q$ is the defined as the difference between the values of the primal and dual problems:

$$g(q) := D^{KL}(q\|p) - w(q) = \max_{s \in \mathrm{conv}(\mathcal{A})} \langle q - s, \log \frac{q}{p} \rangle. \qquad (10)$$

Note that the duality gap is a bound on the primal error as:

$$g(q) = \max_{s \in \mathrm{conv}(\mathcal{A})} \langle q - s, \log \frac{q}{p} \rangle \geq \langle q - q^\star, \log \frac{q}{p} \rangle \geq D^{KL}(q\|p) - D^{KL}(q^\star\|p), \qquad (11)$$

where the first inequality comes from the fact that the optimum $q^\star \in \mathrm{conv}(\mathcal{A})$ and the second from the convexity of the KL divergence w.r.t. $q$. $\qquad\square$

## B  Code Example

Here we include a Python example of how a practitioner would use our method to do Bayesian logistic regression. Just as in the *Edward* framework, the user defines their probabilistic model in terms of the prior w, the input data X, and the likelihood of $y$. Then, qt is initialized. This is ultimately the variational approximation to the posterior and what is returned by our algorithm. Finally, for each iteration t, we create a new variable sw to be optimized by the LMO and then run the black box LMO solver. The user need only specify the model and the base family. The optimization is left up to the black box RELBO solver.

```python
1   import tensorflow as tf
2   import edward as ed
3   from edward.models import Laplace
4   import relbo # our method: boosting BBVI
5
6   # Bayesian logistic regression model
7   w = Normal(loc=tf.zeros(D), scale=1.0 * tf.ones(D)) # Gaussian prior on w
8   X = tf.placeholder(tf.float32, [N, D])
9   y = Bernoulli(logits=ed.dot(X, w))
10
11  # initialize the mixture which represents the latest iterate, qt.
12  qt = Mixture(Laplace)
13
14  # run boosting BBVI 'n_iterations' times
15  # note that at iteration 0, qt is empty, and 'relbo.inference' is performing regular BBVI.
16  for t in range(n_iterations):
17      # Create a new s to be optimized in this iteration
18      loc = tf.get_variable(initializer=tf.random_normal(dims))
19      scale = tf.nn.softplus(tf.get_variable(initializer=tf.random_normal(dims)))
20      sw = Laplace(loc=loc, scale=scale)
21
22      # Run the LMO. Pass in previous iterates to compute RELBO term
23      inference = relbo.KLqp({w: sw}, fw_iterates=qt, data={X: Xtrain, y: ytrain})
24      tf.global_variables_initializer().run()
25      inference.run()
26
27      update_iterates(qt, sw, t)
28
29  return qt
```