[Reviews · NeurIPS 2018]

Reviewer 1



In the submission, the authors aim at developing a black-box boosting method for variational inference, which takes a family of variational distributions and finds a mixture of distribution in a given family that approximates a given posterior distribution well. The main keyword here is black-box; white-box, restricted approaches exist. In order to achieve their aim, the authors formulate a version of the Frank-Wolfe algorithm, and instantiate it with the usual KL objective of variational inference. They then derive a condition on the convergence of this instantiation that is more permissive than the usual smoothness and is based on the reformulation of the bounded curvature condition (Theorem 2). They also show how the constrained optimization problem included in the instantiation of Frank-Wolfe can be expressed in terms of a more intuitive objective, called RELBO in the submission. Finally, the authors propose a criterion of stopping the algorithm (Lemma 4). The resulting black-box algorithm is tested with synthetic data, Bayesian logistic regression, and Bayesian matrix factorization, and these experiments show the promise of the algorithm. I am not an expert on boosting, but I enjoyed reading the submission and found the results interesting. I particularly liked the fact that the RELBO can be understood intuitively. I have a few complaints though, mostly about presentation. Someone like me (who is familiar with variational inference but not boosting nor Frank-Wolfe) might have appreciated the submission more if the authors had explained in more detail what the LMO objective attempts to do intuitively and how the formula at the bottom of page 5 is derived from the definition of LMO in equation (6), and the authors had described how one should implement the stopping criterion. I also struggled to follow the discussion right after Theorem 2, and I couldn't quite see where the delta relaxation in (7) is used in the authors' algorithm. Finally, the held-out test likelihood for Bayesian matrix factorization does not seem to show that Boosted BBVI beats BBVI, but the text (329-330) says it differently. This made me very confused. * equation (3), p3: p(x,z) ===> p(z|x) * What do you mean by L-smooth? * boldface q0 in Algorithm 1 ===> q^0 * second display, p6: D^KL(s || ...Z) ===> D^KL(s || .../Z)

Reviewer 2



The authors consider allowing more flexible variational approximations in black box variational inference, borrowing ideas from boosting/Franke-Wolfe optimization. The key contributions are theoretical: looser conditions for guaranteed convergence are found allowing more straightforward implementation, degenerate solutions are avoided via regularizing the entropy, and a bound on the primal error is used as a principled stopping criterion. Promising empirical results are presented on a) a simple synthetic mixture b) logistic regression c) matrix factorization. The paper is well-written and clearly explained. Having not looked carefully at boosted VI before however it would be helpful if the paper explained how elements in conv(Q) are represented explicitly. I think this is as a mixture of elements in Q, with a new "component" added at every iteration. However this appears not to be the case for the matrix factorization example? The intro on vanilla VI could be compressed to allow for this discussion (I think most people know that material by this point). As a minor point you should specify how the 3 variants in fig 1 correspond to those in algo 1 (I know it's pretty obvious but anyway). The empirical results all seem pretty encouraging and I'm pleased to see the authors used Edward rather than trying to roll their own solution (unless they're Edward people in which case fair enough too). Please do make code available. It would be interesting to see some discussion/quantificaiton of the copmutation/memory/storage requirements as the complexity of the variational posterior grows through the algorithm run. Does this limit applicability to larger models? Overall I felt this paper made solid theoretical contributions and demonstrated their empirical utility sufficently well.

Reviewer 3



The work proposed a new boosting variational inference algorithm, which removes the restrictions of the bounded support and avoid returning the degenerated distributions. In this way, the proposed framework allows users to design any variational family and to use the existing black-box VI software. The algorithm is examined by both simulations and practical applications. This is a piece of solid work. It significantly advances boosting variational inference framework, including relaxing the sufficient conditions of convergence, giving convergence rate, proposing residual ELBO, stopping criterion using duality gap, etc. Can the code be shared?